# Learning Deterministic Weighted Automata with Queries and Counterexamples

**Gail Weiss**
Technion
sgailw@cs.technion.ac.il

**Yoav Goldberg**
Bar Ilan University
Allen Institute for AI
yogo@cs.biu.ac.il

**Eran Yahav**
Technion
yahave@cs.technion.ac.il

## Abstract

We present an algorithm for extraction of a probabilistic *deterministic* finite automaton (PDFA) from a given black-box language model, such as a recurrent neural network (RNN). The algorithm is a variant of the exact-learning algorithm L*, adapted to a probabilistic setting with noise. The key insight is the use of conditional probabilities for observations, and the introduction of a local tolerance when comparing them. When applied to RNNs, our algorithm often achieves better word error rate (WER) and normalised distributed cumulative gain (NDCG) than that achieved by spectral extraction of weighted finite automata (WFA) from the same networks. PDFAs are substantially more expressive than n-grams, and are guaranteed to be stochastic and deterministic – unlike spectrally extracted WFAs.

## 1 Introduction

We address the problem of learning a probabilistic deterministic finite automaton (PDFA) from a trained recurrent neural network (RNN) [17]. RNNs, and in particular their gated variants GRU [13, 14] and LSTM [21], are well known to be very powerful for sequence modelling, but are not interpretable. PDFAs, which explicitly list their states, transitions, and weights, are more interpretable than RNNs [20], while still being analogous to them in behaviour: both emit a single next-token distribution from each state, and have deterministic state transitions given a state and token. They are also much faster to use than RNNs, as their sequence processing does not require matrix operations.

We present an algorithm for reconstructing a PDFA from any given black-box distribution over sequences, such as an RNN trained with a language modelling objective (LM-RNN). The algorithm is applicable for reconstruction of any weighted deterministic finite automaton (WDFA), and is guaranteed to return a PDFA when the target is stochastic – as an LM-RNN is.

***Weighted Finite Automata (WFA)*** A WFA is a weighted *non-deterministic* finite automaton, capable of encoding language models but also other, non-stochastic weighted functions. Ayache et al. [2] and Okudono et al. [24] show how to apply *spectral learning* [5] to an LM-RNN to learn a weighted finite automaton (WFA) approximating its behaviour.

***Probabilistic Deterministic Finite Automata (PDFAs)*** are a weighted variant of DFAs where each state defines a categorical next-token distribution. Processing a sequence in a PDFA is simple: input tokens are processed one by one, getting the next state and probability for each token by table lookup.

WFAs are non-deterministic and so not immediately analogous to RNNs. They are also slower to use than PDFAs, as processing each token in an input sequence requires a matrix multiplication. Finally, spectral learning algorithms are not guaranteed to return stochastic hypotheses even when the target is stochastic – though this can remedied by using quadratic weighted automata [3] and normalising their weights. For these reasons we prefer PDFAs over WFAs for RNN approximation. Formally:

***Problem Definition*** Given an LM-RNN $R$, find a PDFA $W$ approximating $R$, such that for any prefix $p$ its next-token distributions in $W$ and in $R$ have low total variation distance between them.

Existing works on PDFA reconstruction assume a sample based paradigm: the target cannot be queried explicitly for a sequence's probability or conditional probabilities [15, 9, 6]. As such, these methods cannot take full advantage of the information available from an LM-RNN[1]. Meanwhile, most work on the extraction of finite automata from RNNs has focused on "binary" deterministic finite automata (DFAs) [25, 12, 38, 39, 23], which cannot fully express the behaviour of an LM-RNN.

***Our Approach*** Following the successful application of L* [1] to RNNs for DFA extraction [39], we develop an adaptation of L* for the weighted case. The adaptation returns a PDFA when applied to a stochastic target such as an LM-RNN. It interacts with an oracle using two types of queries:

1. *Membership Queries*: requests to give the target probability of the last token in a sequence.
2. *Equivalence Queries*: requests to accept or reject a hypothesis PDFA, returning a *counterexample* — a sequence for which the hypothesis automaton and the target language diverge beyond the tolerance on the next token distribution — if rejecting.

The algorithm alternates between filling an *observation table* with observations of the target behaviour, and presenting minimal PDFAs consistent with that table to the oracle for equivalence checking. This continues until an automaton is accepted. The use of conditional properties in the observation table prevents the observations from vanishing to $0$ on low probabilities. To the best of our knowledge, this is the first work on learning PDFAs from RNNs.

A key insight of our adaptation is the use of an *additive variation tolerance* $t \in [0, 1]$ when comparing rows in the table. In this framework, two probability vectors are considered $t$-equal if their probabilities for each event are within $t$ of each other. Using this tolerance enables us to extract a much smaller PDFA than the original target, while still making locally similar predictions to it on any given sequence. This is necessary because RNN states are real valued vectors, making the potential number of reachable states in an LM-RNN unbounded. The tolerance is non-transitive, making construction of PDFAs from the table more challenging than in L*. Our algorithm suggests a way to address this.

Even with this tolerance, reaching equivalence may take a long time for large target PDFAs, and so we design our algorithm to allow anytime stopping of the extraction. The method allows the extraction to be limited while still maintaining certain guarantees on the reconstructed PDFA.

*Note.* While this paper only discusses RNNs, the algorithm itself is actually agnostic to the underlying structure of the target, and can be applied to any language model. In particular it may be applied to transformers [35, 16]. However, in this case the analogy to PDFAs breaks down.

**Contributions** The main contributions of this paper are:

1. An algorithm for reconstructing a WDFA from any given weighted target, and in particular a PDFA if the target is stochastic.
2. A method for anytime extraction termination while still maintaining correctness guarantees.
3. An implementation of the algorithm [2] and an evaluation over extraction from LM-RNNs, including a comparison to other LM reconstruction techniques.

## 2 Related Work

In Weiss et al [39], we presented a method for applying Angluin's exact learning algorithm L*[1] to RNNs, successfully extracting deterministic finite automata (DFAs) from given binary-classifier RNNs. This work expands on this by adapting L*to extract PDFAs from LM-RNNs. To apply exact learning to RNNs, one must implement equivalence queries: requests to accept or reject a hypothesis. Okudono et al. [24] show how to adapt the equivalence query presented in [39] to the weighted case.

There exist many methods for PDFA learning, originally for acyclic PDFAs [31, 29, 10], and later for PDFAs in general [15, 9, 33, 26, 11, 6]. These methods split and merge states in the learned PDFAs

according to sample-based estimations of their conditional distributions. Unfortunately, they require very large sample sets to succeed (e.g., [15] requires ~13m samples for a PDFA with $|Q|, |\Sigma| = 2$).

Distributions over $\Sigma^*$ can also be represented by WFAs, though these are non-deterministic. These can be learned using *spectral algorithms*, which use SVD decomposition and $|\Sigma| + 1$ matrices of observations from the target to build a WFA [4, 5, 8, 22]. Spectral algorithms have recently been applied to RNNs to extract WFAs representing their behaviour [2, 24, 28], we compare to [2] in this work. The choice of observations used is also a focus of research in this field [27].

For more on language modelling, see the reviews of Goodman [19] or Rosenfeld [30], or the Sequence Prediction Challenge (SPiCe) [7] and Probabilistic Automaton Challenge (PAutomaC) [36].

# 3  Background

***Sequences and Notations*** For a finite alphabet $\Sigma$, the set of finite sequences over $\Sigma$ is denoted by $\Sigma^*$, and the empty sequence by $\varepsilon$. For any $\Sigma$ and stopping symbol $\$ \notin \Sigma$, we denote $\Sigma_\$ \triangleq \Sigma \cup \{\$\}$, and $\Sigma^{+\$} \triangleq \Sigma^* \cdot \Sigma_\$$ – the set of $s \in \Sigma_\$ \setminus \{\varepsilon\}$ where the stopping symbol may only appear at the end.

For a sequence $w \in \Sigma^*$, its length is denoted $|w|$, its concatenation after another sequence $u$ is denoted $u \cdot w$, its $i$-th element is denoted $w_i$, and its prefix of length $k \leq |w|$ is denoted $w_{:k} = w_1 \cdot ... \cdot w_k$. We use the shorthand $w_{-1} \triangleq w_{|w|}$ and $w_{:-1} \triangleq w_{:|w|-1}$. A set of sequences $S \subseteq \Sigma^*$ is said to be *prefix closed* if for every $w \in S$ and $k \leq |w|$, $w^k \in S$. *Suffix closedness* is defined analogously.

For any finite alphabet $\Sigma$ and set of sequences $S \subseteq \Sigma^*$, we assume some internal ordering of the set's elements $s_1, s_2, ...$ to allow discussion of vectors of observations over those elements.

***Probabilistic Deterministic Finite Automata (PDFAs)*** are tuples $A = \langle Q, \Sigma, \delta_Q, q^i, \delta_W \rangle$ such that $Q$ is a finite set of states, $q^i \in Q$ is the initial state, $\Sigma$ is the finite input alphabet, $\delta_Q : Q \times \Sigma \to Q$ is the transition function and $\delta_W : Q \times \Sigma_\$ \to [0, 1]$ is the transition weight function, satisfying $\sum_{\sigma \in \Sigma_\$} \delta_W(q, \sigma) = 1$ for every $q \in Q$.

The recurrent application of $\delta_Q$ to a sequence is denoted by $\hat{\delta} : Q \times \Sigma^* \to Q$, and defined: $\hat{\delta}(q, \varepsilon) \triangleq q$ and $\hat{\delta}(q, w \cdot a) \triangleq \delta_Q(\hat{\delta}(q, w), a)$ for every $q \in Q, a \in \Sigma, w \in \Sigma^*$. We abuse notation to denote: $\hat{\delta}(w) \triangleq \hat{\delta}(q^i, w)$ for every $w \in \Sigma^*$. If for every $q \in Q$ there exists a series of non-zero transitions reaching a state $q$ with $\delta_W(q, \$) > 0$, then $A$ defines a distribution $P_A$ over $\Sigma^*$ as follows: for every $w \in \Sigma^*$, $P_A(w) = \delta_W(\hat{\delta}(w), \$) \cdot \prod_{i \leq |w|} \delta_W(\hat{\delta}(w_{:i-1}), w_i)$.

***Language Models (LMs)*** Given a finite alphabet $\Sigma$, a *language model* $M$ over $\Sigma$ is a model defining a distribution $P_M$ over $\Sigma^*$. For any $w \in \Sigma^*, S \subset \Sigma^{+\$}$, and $\sigma \in \Sigma$, $P = P_M$ induces the following:

- *Prefix Probability:* $P^p(w) \triangleq \sum_{v \in \Sigma^*} P(w \cdot v)$.
- *Last Token Probability:* if $P^p(w) > 0$, then $P^l(w \cdot \sigma) \triangleq \frac{P^p(w \cdot \sigma)}{P^p(w)}$ and $P^l(w \cdot \$) \triangleq \frac{P(w)}{P^p(w)}$.
- *Last Token Probabilities Vector:* if $P^p(w) > 0$, $P_S^l(w) \triangleq (P^l(w \cdot s_1), ..., P^l(w \cdot s_{|S|}))$.
- *Next Token Distribution:* $P^n(w) : \Sigma_\$ \to [0, 1]$, defined: $P^n(w)(\sigma) = P^l(w \cdot \sigma)$.

***Variation Tolerance*** Given two categorical distributions $\mathbf{p}$ and $\mathbf{q}$, their total variation distance is defined $\delta(\mathbf{p}, \mathbf{q}) \triangleq \|\mathbf{p} - \mathbf{q}\|_\infty$, i.e., the largest difference in probabilities that they assign to the same event. Our algorithm tolerates some variation distance between next-token probabilities, as follows:

Two event probabilities $p_1, p_2$ are called *t-equal* and denoted $p_1 \approx_t p_2$ if $|p_1 - p_2| \leq t$. Similarly, two vectors of probabilities $\mathbf{v_1}, \mathbf{v_2} \in [0, 1]^n$ are called *t-equal* and denoted $\mathbf{v_1} \approx_t \mathbf{v_2}$ if $\|\mathbf{v_1} - \mathbf{v_2}\|_\infty \leq t$, i.e. if $\max_{i \in [n]}(|\mathbf{v_{1_i}} - \mathbf{v_{2_i}}|) \leq t$. For any distribution $P$ over $\Sigma^*$, $S \subset \Sigma^{+\$}$, and $p_1, p_2 \in \Sigma^*$, we denote $p_1 \approx_{(P,S,t)} p_2$ if $P_S^l(p_1) \approx_t P_S^l(p_2)$, or simply $p_1 \approx_{(S,t)} p_2$ if $P$ is clear from context. For any two language models $A, B$ over $\Sigma^*$ and $w \in \Sigma^{+\$}$, we say that $A, B$ are *t-consistent on $w$* if $P_A^l(u) \approx_t P_B^l(u)$ for every prefix $u \neq \varepsilon$ of $w$. We call $t$ the *variation tolerance*.

***Oracles and Observation Tables*** Given an oracle $\mathcal{O}$, an observation table for $\mathcal{O}$ is a sequence indexed matrix $O_{P,S}$ of observations taken from it, with the rows indexed by prefixes $P$ and the columns

by suffixes $S$. The observations are $O_{P,S}(p,s) = \mathcal{O}(p{\cdot}s)$ for every $p \in P$, $s \in S$. For any $p \in \Sigma^*$ we denote $\mathcal{O}_S(p) \triangleq (\mathcal{O}(p{\cdot}s_1), ..., \mathcal{O}(p{\cdot}s_2))$, and for every $p \in P$ the $p$-th row in $O_{P,S}$ is denoted $O_{P,S}(p) \triangleq \mathcal{O}_S(p)$. In this work we use an oracle for the last-token probabilities of the target, $\mathcal{O}(w) = P^l(w)$ for every $w \in \Sigma^{+\$}$, and maintain $S \subseteq \Sigma^{+\$}$.

***Recurrent Neural Networks (RNNs)*** An RNN is a recursive parametrised function $h_t = f(x_t, h_{t-1})$ with initial state $h_0$, such that $h_t \in \mathbb{R}^n$ is the state after time $t$ and $x_t \in X$ is the input at time $t$. A language model RNN (LM-RNN) over an alphabet $X = \Sigma$ is an RNN coupled with a prediction function $g : h \mapsto d$, where $d \in [0,1]^{|\Sigma_\$|}$ is a vector representation of a next-token distribution. RNNs differ from PDFAs only in that their number of reachable states (and so number of different next-token distributions for sequences) may be unbounded.

## 4  Learning PDFAs with Queries and Counterexamples

In this section we describe the details of our algorithm. We explain why a direct application of L\* to PDFAs will not work, and then present our non-trivial adaptation. Our adaptation does not rely on the target being stochastic, and can in fact be applied to reconstruct any WDFA from an oracle.

***Direct application of L\* does not work for LM-RNNs:*** L\* is a polynomial-time algorithm for learning a deterministic finite automaton (DFA) from an oracle. It can be adapted to work with oracles giving any finite number of classifications to sequences, and can be naively adapted to a probabilistic target $P$ with finite possible next-token distributions $\{P^n(w)|w \in \Sigma^*\}$ by treating each next-token distribution as a sequence classification. However, *this will not work for reconstruction from RNNs*. This is because the set of reachable states in a given RNN is unbounded, and so also the set of next-token distributions. Thus, in order to practically adapt L\* to extract PDFAs from LM-RNNs, we must reduce the number of classes L\* deals with.

***Variation Tolerance*** Our algorithm reduces the number of classes it considers by allowing an additive variation tolerance $t \in [0,1]$, and considering $t$-equality (as presented in Section 3) as opposed to actual equality when comparing probabilities. In introducing this tolerance we must handle the fact that it may be non-transitive: there may exist $a,b,c \in [0,1]$ such that $a \approx_t b, b \approx_t c$, but $a \not\approx_t c$. [3]

To avoid potentially grouping together all predictions on long sequences, which are likely to have very low probabilities, our algorithm observes only local probabilities. In particular, the algorithm uses an oracle that gives the last-token probability for every non-empty input sequence.

### 4.1  The Algorithm

The algorithm loops over three main steps: (1) expanding an observation table $O_{P,S}$ until it is closed and consistent, (2) constructing a hypothesis automaton, and (3) making an equivalence query about the hypothesis. The loop repeats as long as the oracle returns counterexamples for the hypotheses. In our setting, counterexamples are sequences $w \in \Sigma^*$ after which the hypothesis and the target have next-token distributions that are not $t$-equal. They are handled by adding all of their prefixes to $P$.

Our algorithm expects last token probabilities from the oracle, i.e.: $\mathcal{O}(w) = P_T^l(w)$ where $P_T$ is the target distribution. The oracle is not queried on $P_T^l(\varepsilon)$, which is undefined. To observe the entirety of every prefix's next-token distribution, $O_{P,S}$ is initiated with $P = \{\varepsilon\}, S = \Sigma_\$$.

**Step 1: Expanding the observation table**   $O_{P,S}$ is expanded as in L\* [1], but with the definition of row equality relaxed. Precisely, it is expanded until:

1. *Closedness* For every $p_1 \in P$ and $\sigma \in \Sigma$, there exists some $p_2 \in P$ such that $p_1{\cdot}\sigma \approx_{S,t} p_2$.
2. *Consistency* For every $p_1, p_2 \in P$ such that $p_1 \approx_{S,t} p_2$, for every $\sigma \in \Sigma$, $p_1{\cdot}\sigma \approx_{S,t} p_2{\cdot}\sigma$.

The table expansion is managed by a queue $L$ initiated to $P$, from which prefixes $p$ are processed one at a time as follows: If $p \notin P$, and there is no $p' \in P$ s.t. $p \approx_{(t,S)} p'$, then $p$ is added to $P$. If $p \in P$ already, then it is checked for inconsistency, i.e. whether there exist $p', \sigma$ s.t. $p \approx_{(t,S)} p'$ but

$p \cdot \sigma \not\approx_{(t,S)} p' \cdot \sigma$. In this case a *separating suffix* $\tilde{s}$, $P_T^l(p \cdot \sigma \cdot \tilde{s}) \not\approx_t P_T^l(p' \cdot \sigma \cdot \tilde{s})$ is added to $S$, such that now $p \not\approx_{t,S} p'$, and the expansion restarts. Finally, if $p \in P$ then $L$ is updated with $p \cdot \Sigma$.

As in L*, checking closedness and consistency can be done in arbitrary order. However, if the algorithm may be terminated before $O_{P,S}$ is closed and consistent, it is better to process $L$ in order of prefix probability (see section 4.2).

**Step 2: PDFA construction**    Intuitively, we would like to group equivalent rows of the observation table to form the states of the PDFA, and map transitions between these groups according to the table's observations. The challenge in the variation-tolerating setting is that *t-equality is not transitive*.

Formally, let $C$ be a partitioning (*clustering*) of $P$, and for each $p \in P$ let $c(p) \in C$ be the partition (*cluster*) containing $p$. $C$ should satisfy:

1. *Determinism* For every $c \in C, p_1, p_2 \in c, \sigma \in \Sigma$: $p_1 \cdot \sigma, p_2 \cdot \sigma \in P \implies c(p_1 \cdot \sigma) = c(p_2 \cdot \sigma)$.
2. *t-equality (Cliques)* For every $c \in C$ and $p_1, p_2 \in c$, $p_1 \approx_{(t,S)} p_2$.

For $c \in C, \sigma \in \Sigma$, we denote $C_{c,\sigma} = \{c(p \cdot \sigma) | p \in c, p \cdot \sigma \in P\}$ the next-clusters reached from $c$ with $\sigma$, and $k_{c,\sigma} \triangleq |C_{c,\sigma}|$. Note that $C$ satisfies determinism iff $k_{c,\sigma} \leq 1$ for every $c \in C, \sigma \in \Sigma$. Note also that the constraints are always satisfiable by the clustering $C = \{\{p\}\}_{p \in P}$

We present a 4-step algorithm to solve these constraints while trying to avoid excessive partitions: [4]

1. *Initialisation*: The prefixes $p \in P$ are partitioned into some initial clustering $C$ according to the $t$-equality of their rows, $O_S(p)$.
2. *Determinism I*: $C$ is refined until it satisfies determinism: clusters $c \in C$ with tokens $\sigma$ for which $k_{c,\sigma} > 1$ are split by next-cluster equivalence into $k_{c,\sigma}$ new clusters.
3. *Cliques*: Each cluster is refined into cliques (with respect to $t$-equality).
4. *Determinism II*: $C$ is again refined until it satisfies determinism, as in (2).

Note that refining a partitioning into cliques may break determinism, but refining into a deterministic partitioning will not break cliques. In addition, when only allowed to refine clusters (and not merge them), all determinism refinements are necessary. Hence the order of the last 3 stages.

Once the clustering $C$ is found, a PDFA $\mathcal{A} = \langle C, \Sigma, \delta_Q, c(\varepsilon), \delta_W \rangle$ is constructed from it. Where possible, $\delta_Q$ is defined directly by $C$: for every $p \cdot \sigma \in P$, $\delta_Q(c(p), \sigma) \triangleq c(p \cdot \sigma)$. For $c, \sigma$ for which $k_{c,\sigma} = 0$, $\delta_Q(c, \sigma)$ is set as the best cluster match for $p \cdot \sigma$, where $p = \text{argmax}_{p \in c} P_T^p(p)$. This is chosen according to the heuristics presented in Section 4.2. The weights $\delta_W$ are defined as follows: for every $c \in C, \sigma \in \Sigma_\$$, $\delta_W(c, \sigma) \triangleq \frac{\sum_{p \in c} P_T^p(p) \cdot P_T^l(p \cdot \sigma)}{\sum_{p \in c} P_T^p(p)}$.

**Step 3: Answering Equivalence Queries**    We sample the target LM-RNN and hypothesis PDFA $\mathcal{A}$ a finite number of times, testing every prefix of each sample to see if it is a counterexample. If none is found, we accept $\mathcal{A}$. Though simple, we find this method to be sufficiently effective in practice. A more sophisticated approach is presented in [24].

## 4.2    Practical Considerations

We present some methods and heuristics that allow a more effective application of the algorithm to large (with respect to $|\Sigma|, |Q|$) or poorly learned grammars.

*Anytime Stopping* In case the algorithm runs for too long, we allow termination before $O_{P,S}$ is closed and consistent, which may be imposed by size or time limits on the table expansion. If $|S|$ reaches its limit, the table expansion continues but stops checking consistency. If the time or $|P|$ limits are reached, the algorithm stops, constructing and accepting a PDFA from the table as is. The construction is unchanged up to the fact that some of the transitions may not have a defined destination, for these we use a "best cluster match" as described in section 4.2. This does not harm the guarantees on $t$-consistency between $O_{P,S}$ and the returned PDFA discussed in Section 5.

*Order of Expansion* As some prefixes will not be added to $P$ under anytime stopping, the order in which rows are checked for closedness and consistency matters. We sort $L$ by prefix weight.

Moreover, if a prefix $p_1$ being considered is found inconsistent w.r.t. some $p_2 \in P, \sigma \in \Sigma_\$$, then all such pairs $p_2, \sigma$ are considered and the separating suffix $\tilde{s} \in \sigma \cdot S$, $\mathcal{O}(p_1 \cdot \tilde{s}) \not\approx_t \mathcal{O}(p_2 \cdot \tilde{s})$ with the highest minimum conditional probability $\max_{p_2} \min_{i=1,2} \frac{P_T^p(p_i \cdot \tilde{s})}{P_T^p(p_i)})$ is added to $S$.

**Best Cluster Match**    Given a prefix $p \notin P$ and set of clusters $C$, we seek a best fit $c \in C$ for $p$. First we filter $C$ for the following qualities until one is non-empty, in order of preference: (1) $c' = c \cup \{p\}$ is a clique w.r.t. $t$-equality. (2) There exists some $p' \in c$ such that $p' \approx_{(t,S)} p$, and $c$ is not a clique. (3) There exists some $p' \in c$ such that $p' \approx_{(t,S)} p$. If no clusters satisfy these qualities, we remain with $C$. From the resulting group $C'$ of potential matches, the best match could be the cluster $c$ minimising $||O_S(p') - O_S(p)||_\infty, p' \in c$. In practice, we choose from $C'$ arbitrarily for efficiency.

**Suffix and Prefix Thresholds**    Occasionally when checking the consistency of two rows $p_1 \approx_t p_2$, a separating suffix $\sigma \cdot s \in \Sigma \cdot S$ will be found that is actually very unlikely to be seen after $p_1$ or $p_2$. In this case it is unproductive to add $\sigma \cdot s$ to $S$. Moreover – especially as RNNs are unlikely to perfectly learn a probability of 0 for some event – it is possible that going through $\sigma \cdot s$ will reach a large number of 'junk' states. Similarly when considering a prefix $p$, if $P_T^l(p)$ is very low then it is possible that it is the failed encoding of probability 0, and that all states reachable through $p$ are not useful.

We introduce thresholds $\varepsilon_S$ and $\varepsilon_P$ for both suffixes and prefixes. When a potential separating suffix $\tilde{s}$ is found from prefixes $p_1$ and $p_2$, it is added to $S$ only if $\min_{i=1,2} P^p(p_i \cdot \tilde{s})/P^p(p_i) \geq \varepsilon_S$. Similarly, potential new rows $p \notin P$ are only added to $P$ if $P^l(p) \geq \varepsilon_P$.

**Finding Close Rows**    We maintain $P$ in a KD-tree $T$ indexed by row entries $O_{P,S}(p)$, with one level for every column $s \in S$. When considering of a prefix $p \cdot \sigma$, we use $T$ to get the subset of all potentially $t$-equal prefixes. $T$'s levels are split into equal-length intervals, we find $2t$ to work well.

**Choosing the Variation Tolerance**    In our initial experiments (on SPiCes 0-3), we used $t = 1/|\Sigma|$. The intuition was that given no data, the fairest distribution over $|\Sigma|$ is the uniform distribution, and so this may also be a reasonable threshold for a significant difference between two probabilities. In practice, we found that $t = 0.1$ often strongly differentiates states even in models with larger alphabets – except for SPiCe 1, where $t = 0.1$ quickly accepted a model of size 1. A reasonable strategy for choosing $t$ is to begin with a large one, and reduce it if equivalence is reached too quickly.

## 5   Guarantees

We note some guarantees on the extracted model's qualities and relation to its target model. *Formal statements and full proofs for each of the guarantees listed here are given in appendix A.*

**Model Qualities**    The model is guaranteed to be deterministic by construction. Moreover, if the target is stochastic, then the returned model is guaranteed to be stochastic as well.

**Reaching Equivalence**    If the algorithm terminates successfully (i.e., having passed an equivalence query), then the returned model is $t$-consistent with the target on every sequence $w \in \Sigma^*$, by definition of the query. In practice we have no true oracle and only approximate equivalence queries by sampling the models, and so can only attain a probable guarantee of their relative $t$-consistency.

$t$-**Consistency and Progress**    No matter when the algorithm is stopped, the returned model is always $t$-consistent with its target on every $p \in P \cdot \Sigma_\$$, where $P$ is the set of prefixes in the table $O_{P,S}$. Moreover, as long as the algorithm is running, the prefix set $P$ is always increased within a finite number of operations. This means that the algorithm maintains a growing set of prefixes on which any PDFA it returns is guaranteed to be $t$-consistent with the target. In particular, this means that if equivalence is not reached, at least *the algorithm's model of the target improves for as long as it runs*.

## 6   Experimental Evaluation

We apply our algorithm to 2-layer LSTMs trained on grammars from the SPiCe competition [7], adaptations of the Tomita grammars [34] to PDFAs, and small PDFAs representing languages with

unbounded history. The LSTMs have input dimensions 2-60 and hidden dimensions 20-100. The LSTMs and their training methods are fully described in Appendix E.

***Compared Methods*** We compare our algorithm to the sample-based method ALERGIA [9], the spectral algorithm used in [2], and $n$-grams. An $n$-gram is a PDFA whose states are a sliding window of length $n-1$ over the input sequence, with transition function $\sigma_1 \cdot ... \cdot \sigma_n, \sigma \mapsto \sigma_2 ... \sigma_n \cdot \sigma$. The probability of a token $\sigma$ from state $s \in \Sigma^{n-1}$ is the MLE estimate $\frac{N(s \cdot \sigma)}{N(s)}$, where $N(w)$ is the number of times the sequence $w$ appears as a subsequence in the samples. For ALERGIA, we use the PDFA/DFA inference toolkit FLEXFRINGE [37].

***Target Languages*** We train 10 RNNs on a subset of the SPiCe grammars, covering languages generated by HMMs, and languages from the NLP, *software*, and *biology* domains. We train 7 RNNs on PDFA adaptations of the 7 Tomita languages [34], made from the minimal DFA for each language by giving each of its states a next-token distribution as a function of whether it is accepting or not. We give a full description of the Tomita adaptations and extraction results in appendix D. As we show in (6.1), the $n$-gram models prove to be very strong competitors on the SPiCe languages. To this end, we consider three additional languages that need to track information for an unbounded history, and thus cannot be captured by *any* $n$-gram model. We call these UHLs (unbounded history languages).

UHLs 1 and 2 are PDFAs that cycle through 9 and 5 states with different next token probabilities. UHL 3 is a weighted adaptation of the 5$^{th}$ Tomita grammar, changing its next-token distribution according to the parity of the seen 0s and 1s. The UHLs are drawn in appendix D.

***Extraction Parameters*** Most of the extraction parameters differ between the RNNs, and are described in the results tables (1, 2). For our algorithm, we always limited the equivalence query to 500 samples. For the spectral algorithm, we made WFAs for all ranks $k \in [50], k = 50m, m \in [10]$, $k = 100m, m \in [10]$, and $k = rank(H)$. For the $n$-grams we used all $n \in [6]$. For these two, we always show the best results for NDCG and WER. For ALERGIA in the FLEXFRINGE toolkit, we use the parameters `symbol_count=50` and `state_count=N`, with N given in the tables.

***Evaluation Measures*** We evaluate the extracted models against their target RNNs on word error rate (WER) and on normalised discounted cumulative gain (NDCG), which was the scoring function for the SPiCe challenge. In particular the SPiCe challenge evaluated models on $NDCG_5$, and we evaluate the models extracted from the SPiCe RNNs on this as well. For the UHLs, we use $NDCG_2$ as they have smaller alphabets. We do not use probabilistic measures such as perplexity, as the spectral algorithm is not guaranteed to return probabilistic automata.

1. *Word error rate (WER)*: The WER of model A against B on a set of predictions is the fraction of next-token predictions (most likely next token) that are different in A and B.
2. *Normalised discounted cumulative gain (NDCG)*: The NDCG of A against B on a set of sequences $\{w\}$ scores A's ranking of the top $k$ most likely tokens after each sequence $w$, $a_1, ..., a_k$, in comparison to the actual most likely tokens given by B, $b_1, ..., b_k$. Formally:

$$NDCG_k(a_1, ..., a_k) = \sum_{n \in [k]} \frac{P_B^l(w \cdot a_n)}{\log_2(n+1)} \Big/ \sum_{n \in [k]} \frac{P_B^l(w \cdot b_n)}{\log_2(n+1)}$$

For NDCG we sample the RNN repeatedly, taking all the prefixes of each sample until we have 2000 prefixes. We then compute the NDCG for each prefix and take the average. For WER, we take 2000 full samples from the RNN, and return the fraction of errors over all of the next-token predictions in those samples. An ideal WER and NDCG is 0 and 1, we note this with $\downarrow, \uparrow$ in the tables.

## 6.1 Results and Discussion

Tables 1 and 2 show the results of extraction from the SPiCe and UHL RNNs, respectively. In them, we list our algorithm as WL*(Weighted L*). For the WFAs and $n$-grams, which are generated with several values of $k$ (rank) and $n$, we show the best scores for each metric. We list the size of the best model for each metric. We do not report the extraction times separately, as they are very similar: the majority of time in these algorithms is spent generating the samples or Hankel matrices.

For PDFAs and WFAs the size columns present the number of states, for the WFAs this is equal to the rank $k$ with which they were reconstructed. For $n$-grams the size is the number of table entries in the model, and the chosen value of $n$ is listed in brackets. In the SPiCe languages, our algorithm did not reach equivalence, and used between 1 and 6 counterexamples for every language before being

| Language ($\lvert\Sigma\rvert, \ell$) | Model | WER↓ | NDCG↑ | Time (h) | WER Size | NDCG Size |
|---|---|---|---|---|---|---|
| SPiCe 0 (4, 1.15) | WL* | 0.084 | 0.987 | 0.3 | 4988 | 4988 |
| | Spectral | **0.053** | **0.996** | 0.3 | k=150 | k=200 |
| | N-Gram | 0.096 | 0.991 | 0.8 | 1118 (n=6) | 1118 (n=6) |
| | ALERGIA | 0.353 | 0.961 | 2.9 | 66 | 66 |
| SPiCe 1 (20, 2.77) | WL*† | **0.093** | **0.971** | 0.4 | 152 | 152 |
| | WL* | 0.376 | 0.891 | 0.1 | 1 | 1 |
| | Spectral | 0.319 | 0.909 | 2.9 | k=12 | k=11 |
| | N-Gram | 0.337 | 0.897 | 0.8 | 8421 (n=4) | 421 (n=3) |
| | ALERGIA | 0.376 | 0.892 | 1.2 | 7 | 7 |
| SPiCe 2 (10, 2.13) | WL*‡ | **0.08** | **0.972** | 0.8 | 962 | 962 |
| | Spectral | 0.263 | 0.893 | 1.6 | k=7 | k=5 |
| | N-Gram | 0.278 | 0.894 | 0.8 | 1111 (n=4) | 1111 (n=4) |
| | ALERGIA | 0.419 | 0.844 | 1.2 | 11 | 11 |
| SPiCe 3 (10, 2.15) | WL*‡ | **0.327** | **0.928** | 1.0 | 675 | 675 |
| | Spectral | 0.466 | 0.843 | 1.2 | k=6 | k=8 |
| | N-Gram | 0.46 | 0.847 | 0.8 | 1111 (n=4) | 11110 (n=5) |
| | ALERGIA ‡‡ | 0.679 | 0.79 | 1.2 | 8 | 8 |
| SPiCe 4 (33, 1.73) | WL* | 0.301 | 0.829 | 0.7 | 4999 | 4999 |
| | Spectral | 0.453 | 0.727 | 1.2 | k=450 | k=250 |
| | N-Gram | **0.099** | **0.968** | 0.8 | 186601 (n=6) | 61851 (n=5) |
| | ALERGIA ‡‡ | 0.639 | 0.646 | 4.4 | 42 | 42 |
| SPiCe 6 (60, 1.66) | WL* | 0.593 | 0.644 | 2.5 | 5000 | 5000 |
| | Spectral | 0.705 | 0.535 | 6.1 | k=17 | k=32 |
| | N-Gram | **0.285** | **0.888** | 0.8 | 127817 (n=5) | 127817 (n=5) |
| | ALERGIA | 0.687 | 0.538 | 1.9 | 26 | 26 |
| SPiCe 7 (20, 1.8) | WL* | 0.626 | 0.642 | 0.5 | 4996 | 4996 |
| | Spectral | 0.801 | 0.472 | 2.4 | k=50 | k=27 |
| | N-Gram | **0.441** | **0.812** | 0.7 | 133026 (n=5) | 133026 (n=5) |
| | ALERGIA | 0.735 | 0.569 | 1.4 | 8 | 8 |
| SPiCe 9 (11, 1.15) | WL* | 0.503 | 0.721 | 0.5 | 4992 | 4992 |
| | Spectral | 0.303 | 0.877 | 1.9 | k=44 | k=44 |
| | N-Gram | **0.123** | **0.961** | 1.0 | 44533 (n=6) | 44533 (n=6) |
| | ALERGIA | 0.501 | 0.739 | 1.1 | 44 | 44 |
| SPiCe 10 (20, 2.1) | WL* | 0.651 | 0.593 | 0.9 | 4987 | 4987 |
| | Spectral | 0.845 | 0.4 | 1.7 | k=42 | k=41 |
| | N-Gram | **0.348** | **0.845** | 0.8 | 153688 (n=5) | 153688 (n=5) |
| | ALERGIA | 0.81 | 0.51 | 2.0 | 13 | 13 |
| SPiCe 14 (27, 0.89) | WL* | 0.442 | 0.716 | 0.8 | 4999 | 4999 |
| | Spectral†† | 0.531 | 0.653 | 2.4 | k=100 | k=100 |
| | N-Gram | **0.079** | **0.977** | 0.7 | 125572 (n=6) | 46158 (n=5) |
| | ALERGIA ‡‡ | 0.641 | 0.611 | 1.2 | 19 | 19 |

Table 1: SPiCe results. Each language is listed with its alphabet size $\lvert\Sigma\rvert$ and RNN test loss $\ell$. The $n$-grams and sample-based PDFAs were created from 5,000,000 samples, and shared samples. FLEXFRINGE was run with state_count=5000. Our algorithm was run with $t$=0.1, $\varepsilon_P$, $\varepsilon_S$=0.01, $\lvert P\rvert\leq$5000 and $\lvert S\rvert\leq$100, and spectral with $\lvert P\rvert, \lvert S\rvert$=1000, with some exceptions: †:$t$=0.05, $\varepsilon_S$, $\varepsilon_P$=0.0, ‡:$\varepsilon_S$=0, ††:$\lvert P\rvert, \lvert S\rvert$=750, ‡‡:state_count=10,000.

stopped – with the exception of SPiCe1 with $t = 0.1$, which reached equivalence on a single state. The UHLs and Tomitas used 0-2 counterexamples each before reaching equivalence.

The SPiCe results show a strong advantage to our algorithm in most of the small synthetic languages (1-3), with the spectral extraction taking a slight lead on SPiCe 0. However, in the remaining SPiCe languages, the $n$-gram strongly outperforms all other methods. Nevertheless, $n$-gram models are inherently restricted to languages that can be captured with bounded histories, and the UHLs demonstrate cases where this property does not hold. Indeed, all the algorithms outperform the $n$-grams on these languages (Table 2).

Our algorithm succeeds in perfectly reconstructing the target PDFA structure for each of the UHL languages, and giving it transition weights within the given variation tolerance (when extracting from the RNN and not directly from the original target, the weights can only be as good as the RNN has learned). The sample-based PDFA learning method, ALERGIA, achieved good WER and NDCG

| Language ($|\Sigma|, \ell$) | Model | WER↓ | NDCG↑ | Time (s) | WER Size | NDCG Size |
|---|---|---|---|---|---|---|
| UHL 1 (2, 0.72) | WL* | **0.0** | **1.0** | 15 | 9 | 9 |
| | Spectral | **0.0** | **1.0** | 56 | k=80 | k=150 |
| | N-Gram | 0.129 | 0.966 | 259 | 63 (n=6) | 63 (n=6) |
| | ALERGIA | 0.004 | 0.999 | 278 | 56 | 56 |
| UHL 2 (5, 1.32) | WL* | **0.0** | **1.0** | 73 | 5 | 5 |
| | Spectral | 0.002 | 1.0 | 126 | k=49 | k=47 |
| | N-Gram | 0.12 | 0.94 | 269 | 3859 (n=6) | 3859 (n=6) |
| | ALERGIA | 0.023 | 0.979 | 329 | 25 | 25 |
| UHL 3 (2, 0.86) | WL* | **0.0** | **1.0** | 55 | 4 | 4 |
| | Spectral | **0.0** | **1.0** | 71 | k=44 | k=17 |
| | N-Gram | 0.189 | 0.991 | 268 | 63 (n=6) | 63 (n=6) |
| | ALERGIA | 0.02 | 0.999 | 319 | 47 | 47 |

Table 2: UHL results. Each language is listed with its alphabet size $|\Sigma|$ and RNN test loss $\ell$. The $n$-grams and sample-based PDFAs were created from 500,000 samples, and shared samples. FLEXFRINGE was run with state_count = 50 . Our algorithm was run with $t=0.1, \varepsilon_P, \varepsilon_S=0.01, |P|\leq5000$ and $|S|\leq100$, and spectral with $|P|, |S|=250$.

scores but did not manage to reconstruct the original PDFA structure. This may be improved by taking a larger sample size, though it comes at the cost of efficiency.

***Tomita Grammars*** The full results for the Tomita extractions are given in Appendix D. All of the methods reconstruct them with perfect or near-perfect WER and NDCG, except for $n$-gram which sometimes fails. For each of the Tomita RNNs, our algorithm extracted and accepted a PDFA with identical structure to the original target in approximately 1 minute (the majority of this time was spent on sampling the RNN and hypothesis before accepting the equivalence query). These PDFAs had transition weights within the variation tolerance of the corresponding target transition weights.

**On the effectiveness of n-grams** The n-gram models prove to be a very strong competitors for many of the languages. Indeed, n-gram models are very effective for learning in cases where the underlying languages have strong local properties, or can be well approximated using local properties, which is rather common (see e.g., Sharan et al. [32]). However, there are many languages, including ones that can be modeled with PDFAs, for which the locality property does not hold, as demonstrated by the UHL experiments.

As $n$-grams are merely tables of observed samples, they are very quick to create. However, their simplicity also works against them: the table grows exponentially in $n$ and polynomially in $|\Sigma|$. In the future, we hope that our algorithm can serve as a base for creating reasonably sized finite state machines that will be competitive on real world tasks.

## 7 Conclusions

We present a novel technique for learning a distribution over sequences from a trained LM-RNN. The technique allows for some variation between the predictions of the RNN's internal states while still merging them, enabling extraction of a PDFA with fewer states than in the target RNN. It can also be terminated before completing, while still maintaining guarantees of local similarity to the target. The technique does not make assumptions about the target model's representation, and can be applied to any language model – including LM-RNNs and transformers. It also does not require a probabilistic target, and can be directly applied to recreate any WDFA.

When applied to stochastic models such as LM-RNNs, the algorithm returns PDFAs, which are a desirable model for LM-RNN extraction because they are deterministic and therefore faster and more interpretable than WFAs. We apply it to RNNs trained on data taken from small PDFAs and HMMs, evaluating the extracted PDFAs against their target LM-RNNs and comparing to extracted WFAs and n-grams. When the LM-RNN has been trained on a small target PDFA, the algorithm successfully reconstructs a PDFA that has identical structure to the target, and local probabilities within tolerance of the target. For simple languages, our method is generally the strongest of all those considered. However for natural languages $n$-grams maintain a strong advantage. Improving our method to be competitive on naturally occuring languages as well is an interesting direction for future work.

## Acknowledgments

The authors wish to thank Rémi Eyraud for his helpful discussions and comments, and Chris Hammerschmidt for his assistance in obtaining the results with FLEXFRINGE . The research leading to the results presented in this paper is supported by the Israeli Science Foundation (grant No.1319/16), and the European Research Council (ERC) under the European Union's Seventh Framework Programme (FP7-2007-2013), under grant agreement no. 802774 (iEXTRACT).

## Footnotes

[1]It is possible to adapt these methods to an active learning setting, in which they may query an oracle for exact probabilities. However, this raises other questions: on which suffixes are prefixes compared? How does one pool the probabilities of two prefixes when merging them? We leave such an adaptation to future work.

[2]Available at `www.github.com/tech-srl/weighted_lstar`

[3]We could define a variation tolerance by quantisation of the distribution space, which would be transitive. However this may be unnecessarily aggressive at the edges of the intervals.

[4]We describe our implementation of these stages in appendix C.

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
