[Supplementary Material]

# Supplementary Material

## A Guarantees

We show that our algorithm returns a PDFA, and discuss the relation between the obtained PDFA $\mathcal{A}$ and the target $T$ when anytime stopping is and isn't used.

### A.1 Probability

**Theorem A.1.** *The algorithm returns a PDFA.*

*Proof.* Let $C$ be the final clustering of $P$ achieved by the method in section 4.1. By construction, the algorithm returns a finite state machine $A = \langle C, \Sigma, c(\varepsilon), \delta_Q, \delta_W, \beta \rangle$ with well defined states, initial state, transition weights and stopping weights. We show that this machine is deterministic and probabilistic, i.e.:

1. *Deterministic*: for every $c \in C, \sigma \in \Sigma$, $\delta_Q(c, \sigma)$ is uniquely defined
2. *Probabilistic*: for every $c \in C, \sigma \in \Sigma$: $\delta_Q(c, \sigma) \in [0, 1]$, $\beta(c) \in [0, 1]$, and $\beta(c) + \sum_{\sigma \in \Sigma} \delta_W(c, \sigma) = 1$.

*Proof of (1):* By the final refinement of the clustering (Determinism II), $k_{c,\sigma} \leq 1$ and so by construction $\delta_Q(c, \sigma)$ is assigned at most one value. If, and only if, $k_{c,\sigma} < 1$, then $\delta_Q(c, \sigma)$ is assigned some best available value. So $\delta_Q(c, \sigma)$ is always assigned exactly one value.

*Proof of (2)*: the values of $\delta_W$ and $\beta$ are weighted averages of probabilities, and so also in $[0, 1]$ themselves. They also sum to 1 as they are averages of distributions. Formally, for every $c \in C$:

$$\beta(c) + \sum_{\sigma \in \Sigma} \delta_W(c, \sigma) =$$

$$\frac{\sum_{p \in c} P_T^p(p) P_T^l(p \cdot \$)}{\sum_{p \in c} P_T^p(p)} + \sum_{\sigma \in \Sigma} \frac{\sum_{p \in c} P_T^p(p) P_T^l(p \cdot \sigma)}{\sum_{p \in c} P_T^p(p)} =$$

$$\frac{\sum_{p \in c} P_T^p(p) P_T^l(p \cdot \$)}{\sum_{p \in c} P_T^p(p)} + \frac{\sum_{p \in c} \sum_{\sigma \in \Sigma} P_T^p(p) P_T^l(p \cdot \sigma)}{\sum_{p \in c} P_T^p(p)} =$$

$$\frac{\sum_{p \in c} P_T^p(p) \sum_{\sigma \in \Sigma_\$} P_T^l(p \cdot \sigma)}{\sum_{p \in c} P_T^p(p)} \underset{(*)}{=} \frac{\sum_{p \in c} P_T^p(p)}{\sum_{p \in c} P_T^p(p)} = 1$$

where $(*)$ follows from the probabilistic behaviour of $T$: $\sum_{\sigma \in \Sigma_\$} P_T^l(p \cdot \sigma) = 1$ for any $p \in \Sigma^*$. □

### A.2 Progress

We consider extraction using noise tolerance $t$ from some target $T = \langle Q, \Sigma, q^i, \delta_Q, \delta_W^T \rangle$. For the observation table $O_{P,S}$ at any stage, we denote $n_{P,S}$ the size of the largest set of pairwise $t$-distinguishable rows $O_S(p), p \in P$.

Let $A$ be an automaton constructed by the algorithm, whether or not it was stopped ahead of time. Let $O_{P,S}$ be the observation table reached before making $A$, $C \subset \mathbb{P}(P)$ be the clustering of $P$ attained when building $A$ from $O_{P,S}$ (i.e., the states of $A$), and denote $A = \langle C, \Sigma, c^i, \delta_C, \delta_W^A \rangle$. Denote $c : P \to C$ the cluster for each prefix, i.e. $p \in c(p)$ for every $p \in P$. In addition, for every cluster $c \in C$, denote $p_c$ the prefix $p_c \in c$ from which $\delta_W^A(c, \circ)$ was defined when building $A$.

We show that as the algorithm progresses, it defines a monotonically increasing group of sequences $W \subset \Sigma^{+\$}$ on which the target $T$ and the algorithm's automata $A$ are $t$-consistent, and that this group is $P \cdot \Sigma_\$$.

**Lemma A.2.** *$P$ is always prefix closed.*

*Proof.* $P$ begins as $\{\varepsilon\}$, which is prefix closed. Only two operations add to $P$: closedness and counterexamples. When adding from closedness, the new prefix added to $P$ is of the form $p \cdot \sigma$ for $p \in P, \sigma \in \Sigma$ and so $P$ remains prefix closed. When adding from a counterexample $w$, $w$ is added along with all of its prefixes, and so $P$ remains prefix closed. □

**Lemma A.3.** *For every $p \in P$, $\hat{\delta}_C(c^i, p) = c(p)$, i.e. $p \in \hat{\delta}_C(c^i, p)$.*

*Proof.* We show this by induction on the length of $p$. For $|p| = 0$ i.e. for $\varepsilon$, $\hat{\delta}_C(\varepsilon) = c^i$ by definition of the recursive application of $\delta_C$, and $c^i = c(\varepsilon)$ by construction (in the algorithm). We assume correctness of the lemma for $|p| = n, p \in P$. Consider $p \in P$, $|p| = n + 1$, denote $p = r \cdot \sigma, r \in \Sigma^*, \sigma \in \Sigma$. By the prefix closedness of $P$, $r \in P$, and so by the assumption $\hat{\delta}_C(r) = c(r)$. Now by the definition of $\hat{\delta}_C$, $\hat{\delta}_C(p) = \delta_C(\hat{\delta}_C(r), \sigma) = \delta_C(c(r), \sigma)$. By the construction of $A$, $c(r)$ is defined such that $\delta_C(c(r), \sigma) = c(p \cdot \sigma)$ for every $s \in c(r)$ s.t. $s \cdot \sigma \in P$, and so in particular for $r \in c(r)$, as $r \cdot \sigma = p \in P$). This results in $\hat{\delta}_C(p) = \delta_Q(c(r), \sigma) = c(p)$, as desired. $\square$

**Lemma A.4.** *For every $p \in P$ and $\sigma \in \Sigma_\$$, $\delta_A(c(p), \sigma) \approx_t P_T^l(p \cdot \sigma)$.*

*Proof.* By construction of $A$, in particular by the clique requirement for the clusters of $C$, all of the prefixes $p' \in c(p)$ satisfy $\mathcal{O}_S(p') = O_S(p') \approx_t O_S(p) = \mathcal{O}_S(p)$, and in particular for $\Sigma_\$ \subseteq S$: $\mathcal{O}_{\Sigma_\$}(p') \approx_t \mathcal{O}_{\Sigma_\$}(p)$ (recall that $S$ is initiated to $\Sigma_\$$ and never reduced). $\delta_A(c(p), \sigma)$ is defined as the weighted average of $\mathcal{O}(p' \cdot \sigma)$ for each of these $p' \in c(p)$, and so it is also $t$-equal to $\mathcal{O}(p \cdot \sigma)$ i.e. $P_T^l(p \cdot \sigma)$, as desired. $\square$

**Theorem A.5.** *For every $p \in P, \sigma \in \Sigma_\$$, $A, T$ are $t$-consistent on $p \cdot \sigma$.*

*Proof.* let $u \neq \varepsilon$ be some prefix of $p \cdot \sigma$. Necessarily $v = u_{:-1}$ is some prefix of $p \in P$, and so by the prefix-closedness of $P$ (theorem A.2) $v \in P$. Denote $a = u_{-1} \in \Sigma_\$$. Then

$$P_T^l(u) = P_T^l(v \cdot a) \approx_t \delta_A(c(v), a) = \delta_A(\hat{\delta}C(v), a) = P_A^l(u)$$

where the second and third transitions are justified for $v \in P$ by theorem A.4 and theorem A.3 respectively. This for any prefix $u \neq \varepsilon$ of $p \cdot \sigma$, and so by definition $A, T$ are $t$-consistent on $p \cdot \sigma$ as desired. $\square$

This concludes the proof that $A, T$ are always $t$-consistent on $P \cdot \Sigma_\$$. We now show that the algorithm increases $P \cdot \Sigma_\$$ every finite number of operations, beginning with a direct result from theorem A.5:

**Corollary A.6.** *Every counterexample increases $P$ by at least $1$*

*Proof.* Recall that counterexamples to proposed automata are sequences $w \in \Sigma^{+\$}$ for which $P_T^l(w) \not\approx_t P_A^l(w)$, and that they are handled by adding all their strict prefixes to $P$. Assume by contradiction some counterexample $w \in \Sigma^{+\$}$ for which $P$ does not increase. Then in particular $w_{:-1} \in P$, and by theorem A.5, $P_T^l(w) = P_T^l(w_{:-1} \cdot w_{-1}) \approx_t P_A^l(w_{:-1} \cdot w_{-1}) = P_A^l(w)$, a contradiction. $\square$

**Lemma A.7.** *Always, $|S| \leq \frac{|P| \cdot (|P|-1)}{2} + |\Sigma_\$|$. (i.e., every $O_{P,S}$ can only have had up to $\frac{|P| \cdot (|P|-1)}{2}$ inconsistencies in its making.)*

*Proof.* $S$ is initiated to $\Sigma_\$$, so its initial size is $|\Sigma_\$|$. $S$ is increased only following inconsistencies, cases in which there exist $p_1, p_2 \in P, \sigma \in \Sigma$ s.t. $p_1 \neq p_2 \; O_S(p_1) \approx_t O_S(p_2)$, but $\mathcal{O}_S(p_1) \not\approx_t \mathcal{O}_S(p_2)$. Once some $p_1, p_2 \in P$ cause a suffix $s$ to be added to $S$, by construction of the algorithm, $O_S(p_1) \not\approx_t O_S(p_2)$ for the remainder of the run (as $s \in S$ is a suffix for which $O(p_1, s) \not\approx_t O(p_2, s)$). There are exactly $\frac{|P| \cdot (|P|-1)}{2}$ pairs $p_1 \neq p_2 \in P$ and so that is the maximum number of possible $S$ may have been increased in any run, giving the maximum size $|S| \leq \frac{|P| \cdot (|P|-1)}{2} + |\Sigma_\$|$. $\square$

(Note: If the $t$-equality relation was transitive, it would be possible to obtain a linear bound in the size of $S$. However as it is not, it is possible that a separating suffix may be added to $S$ that separates $p_1$ and $p_2$ while leaving them both $t$-equal to to some other $p_3$.)

**Corollary A.8** (Progress). *For as long as the algorithm runs, it strictly expands a group $\mathbb{C} \subset \Sigma^*$ of sequences on which the automata $A$ it returns is $t$-consistent with its target $T$.*

Figure B.1: Target PDFA $T$

*Proof.* From theorem A.5, $\mathbb{C} = P \times \Sigma_\$$ is a group of sequences on which $A$ is always $t$-consistent with $T$. We show that $\mathbb{C}$ is strictly expanding as the algorithm progresses, i.e. that every finite number of operations, $P$ is increased by at least one sequence.

The algorithm can be split into 4 operations: searching for and handling an unclosed prefix or inconsistency, building (and presenting) a hypothesis PDFA, or handling a counterexample. We show that each one runs in finite time, and that there cannot be infinite operations without increasing $P$.

### *Finite Runtime of the Operations*

*Building $O_{P,S}$*: Finding and handling an unclosed prefix requires a pass over all $P \times \Sigma$, while comparing row values to $P$ – all finite as $P$ is finite (rows are also finite as $S$ is bounded by $P$'s size). Similarly finding and handling inconsistencies requires a pass over rows for all $P^2 \sigma$, also taking finite time.

*Building an Automaton* requires finding a clustering of $P$ satisfying the conditions and then a straightforward mapping of the transitions between these clusters. The clustering is built by one initial clustering (DBSCAN) over the finite set $P$ and then only refinement operations (without merges). As putting each prefix in its own cluster is a solution to the conditions, a satisfying clustering will be reached in finite time. *Counterexamples* Handling a counterexample $w$ requires adding at most $|w|$ new rows to $O_{P,S}$. As $S$ is finite, this is a finite operation.

**Finite Operations between Additions to $P$** Handling an unclosed prefix by construction increases $P$, and as shown in theorem A.6, so does handling a counterexample. Building a hypothesis is followed by an equivalence query, after which the algorithm will either terminate or a counterexample will be returned (increasing $P$). Finally, by A.7, the number of inconsistencies between every increase of $P$ is bounded. □

## B Example

We extract from the PDFA $T$ presented in B.1 using prefix and suffix thresholds $\varepsilon_P, \varepsilon_S = 0$ and variation tolerance $t = 0.1$. We limit the number of samples per equivalence query to $500$. This extraction will demonstrate both types of table expansions, both types of clustering refinements, and counterexamples. Notice that in our example, the state $q5$ is $t$-equal with respect to next-token distribution to both $q1$ and $q3$, though they themselves are not $t$-equal to each other.

Extraction begins by initiating the table with $P = \{\varepsilon\}, S = \Sigma_\$$, and the queue $Q$ with $P$. We will pop from the queue in order of prefix weight, though this is not necessary when not considering anytime stopping. At this point the table is:

| S / P | a | b | $ |
|---|---|---|---|
| $\varepsilon$ | 0.5 | 0.4 | 0.1 |

Figure B.2: Hypotheses during extraction from $T$

The first prefix considered is $\varepsilon$, it is already in $P$. It is consistent simply as it is not similar to any other $p \in P$. However it might not be closed. Its continuations $\varepsilon \cdot \Sigma = \{a, b\}$ are added to $Q$, to check its closedness later. $Q$ is now $\{a, b\}$.

Next is $a$ (which has prefix weight 0.5). $\mathcal{O}_S(a) = (0.7, 0.25, 0.05)$, which is not $t$-equal to the only row in the table: $O_S(\varepsilon) = (0.5, 0.4, 0.1)$. It follows that $a_{:-1} = \varepsilon$ was not closed, and $a$ is added to $P$. The table is now:

| S<br>P | a | b | $ |
|---|---|---|---|
| $\varepsilon$ | 0.5 | 0.4 | 0.1 |
| a | 0.7 | 0.25 | 0.05 |

$a$ is also consistent simply as it has no $t$-equal rows. Its continuations $a \cdot \Sigma$ are added to $Q$ to check closedness, giving $Q = \{b, ab, aa\}$.

Now for each of $q \in Q$, $\mathcal{O}_S(q) = O_S(\varepsilon)$, meaning that the table is closed. None of the prefixes in $Q$ are added to $P$, and so they are also not checked for consistency. The expansion stops and a clustering $C = \{\{\varepsilon\}, \{a\}\}$ is made ($\varepsilon$ and $a$ are not $t$-equal). The transitions are mapped and the automaton $H1$ shown in figure B.2 is presented for an equivalence query.

$H1$ and $T$ are each sampled according to their distributions up to 500 times, and $P_T^n(p), P_{H1}^n(p)$ are compared for every prefix $p$ of each sample. This soon yields the counterexample $c = aaa$, for which $P_{H1}^n(c) = (0.7, 0.25, 0.05) \not\approx_{0.1} (0.5, 0.4, 0.1) = P_T^n(c)$. $c$'s prefixes $\varepsilon, a, aa, aaa$ are added to $P$ and the expansion restarts with $Q = P$ and table:

| S<br>P | a | b | $ |
|---|---|---|---|
| $\varepsilon$ | 0.5 | 0.4 | 0.1 |
| a | 0.7 | 0.25 | 0.05 |
| aa | 0.5 | 0.4 | 0.1 |
| aaa | 0.5 | 0.4 | 0.1 |

$Q$ is processed: $\varepsilon$ is already in $P$, $a, b$ are added to $Q$. We check its consistency with each of its $t$-equal rows, $aa$ and $aaa$, beginning with $aa$. For $a \in \Sigma$, $\mathcal{O}_S(\varepsilon \cdot a) = (0.7, 0.25, 0.05) \not\approx_{0.1} (0.5, 0.4, 0.1) = \mathcal{O}_S(aa \cdot a)$, with the biggest difference (0.2) being on the suffix $a \in S$. The separating suffix $a \cdot a \in \Sigma \cdot S$ is added to $S$, separating $\varepsilon$ and $aa$ in the table:

| S ＼ P | a | b | $ | aa |
|---|---|---|---|---|
| $\varepsilon$ | 0.5 | 0.4 | 0.1 | 0.7 |
| a | 0.7 | 0.25 | 0.05 | 0.5 |
| aa | 0.5 | 0.4 | 0.1 | 0.5 |
| aaa | 0.5 | 0.4 | 0.1 | 0.6 |

The expansion is restarted with $Q = P$. Eventually all of $P \cdot \Sigma$ are processed and the table is found closed and consistent. The extraction moves to constructing a hypothesis.

An initial clustering is made, in our case using `sklearn.cluster.DBSCAN` with parameter `min_samples=1`. It returns $C_0 = \{\{\varepsilon, aa, aaa\}, \{a\}\}$. However, this does not satisfy the determinism requirement: for $\varepsilon$ and $aa$, which are both in the same cluster, their continuations with $a \in \Sigma$ are also in $P$ and appear in different clusters. The cluster $\{\varepsilon, aa, aaa\}$ is split such that $\varepsilon$ and $aa$ are separated. For $aaa$, whose continuation $aaaa$ is not in $P$, it is not important whether it joins $\varepsilon$ or $aa$, and it is equally close (with respect to $L_\infty$ distance on rows) to both. The new clustering $C = \{\{aa, aaa\}, \{a\}, \{\varepsilon\}\}$ is returned. This clustering satisfies $t$-equality ($aa \approx_{t,S} aaa$), and a hypothesis can be made.

For each cluster $c \in C$ there is a $p \in c$ for which $p \cdot a \in P$ and so all of the $a$-transitions are simple to map. For $b$, the transitions are mapped according to the closest rows in the table, e.g. the $b$-transition from the initial state $c(\varepsilon)$ maps to $c(aa)$, as $\mathcal{O}_S(b) = (0.5, 0.4, 0.1, 0.5) \approx_t (0.5, 0.4, 0.1, 0.5) = O_S(aa)$. This yields the PDFA $H2$ shown in B.2.

Sampling $H2$ and $T$ soon yields the counterexample $bb$, for which $P_T^n(bb) = (0.7, 0.25, 0.05) \not\approx_t (0.5, 0.4, 0.1) = P_{H2}^n(bb)$. All of $bb$'s prefixes are added to $P$, the queue is again initiated to $P$, and expansion restarts with the table:

| S ＼ P | a | b | $ | aa |
|---|---|---|---|---|
| $\varepsilon$ | 0.5 | 0.4 | 0.1 | 0.7 |
| a | 0.7 | 0.25 | 0.05 | 0.5 |
| aa | 0.5 | 0.4 | 0.1 | 0.5 |
| aaa | 0.5 | 0.4 | 0.1 | 0.6 |
| b | 0.5 | 0.4 | 0.1 | 0.5 |
| bb | 0.7 | 0.25 | 0.05 | 0.5 |

When the prefix $b$ is processed, an inconsistency is found: $b \approx_{t,S} aa$, but $\mathcal{O}_S(bb) = (0.7, 0.25, 0.05, 0.5) \not\approx_t (0.5, 0.4, 0.1, 0.6) = \mathcal{O}_S(aab)$, in particular on $a \in S$. $ba$ is added to $S$, $Q$ is reset to $P$, and the expansion restarts with the table:

| S ＼ P | a | b | $ | aa | ba |
|---|---|---|---|---|---|
| $\varepsilon$ | 0.5 | 0.4 | 0.1 | 0.7 | 0.5 |
| a | 0.7 | 0.25 | 0.05 | 0.5 | 0.5 |
| aa | 0.5 | 0.4 | 0.1 | 0.5 | 0.5 |
| aaa | 0.5 | 0.4 | 0.1 | 0.6 | 0.6 |
| b | 0.5 | 0.4 | 0.1 | 0.5 | 0.7 |
| bb | 0.7 | 0.25 | 0.05 | 0.5 | 0.5 |

This time the table is found to be closed and consistent. `DBSCAN` gives the initial clustering $C_0 = \{\{\varepsilon, aa, aaa, b\}, \{a, bb\}\}$, and as before the determinism refinement separates $a$ and $\varepsilon$, giving $C_1 = \{\{aa, aaa, b\}, \{a, bb\}, \{\varepsilon\}\}$. Now the $t$-equality requirement is checked, and the first cluster does not satisfy it: while $aa \approx_{t,S} aaa$ and $b \approx_{t,S} aaa$, $aa \not\approx_{t,S} b$. The cluster is split across the suffix with the largest range, $ba$, yielding the new clustering $C = \{\{aa, aaa\}, \{a, bb\}, \{\varepsilon\}, \{b\}\}$. This clustering satisfies both determinism and $t$-equality and the hypothesis $H3$ is made, with $\sigma$-transitions from clusters $c$ for which there is no $p \in c$ such that $p \cdot \sigma \in P$ (e.g. $b$ from $\{aa, aaa\}$) being made according to closest rows as described before.

Sampling 500 times from each of $H3$ and $T$ yields no counterexample, and indeed none exists even though the two are not exactly the same: the distributions of states $q5, q4$ and $q3$ of $T$ are $t = 0.1$-equal, and the PDFAs $H3$ and $T$ are $t$-equal.

***A note on prefix and suffix thresholds.*** Suppose that instead of $T$, we had a PDFA $T'$ over $\Sigma = \{a, b, c\}$ as follows: $T'$ is identical to $T$, except that from every state $q \in Q_T$ there is a $c$-transition with a very small probability $\varepsilon$ leading to a different state of an extremely large PDFA $L$. If $\varepsilon$ is very small, developing $L$ will be of little benefit for the approximation, but waste a lot of time and space for the extraction. However, if $\varepsilon_S, \varepsilon_P > \varepsilon$, then no prefix containing $c$ will ever be added to the table, and similarly no suffix containing $c$ will ever be considered a separating suffix (needlessly separating two prefixes). The existence of such transitions is quite possible in RNNs: they are unlikely to perfectly learn to represent $0$ even for tokens that have never been seen, and moreover never 'tame' the states that would be reached from such transitions (as they are not seen in training).

## C   Implementation

***Clustering the Prefixes*** The initial clustering can be done with any clustering algorithm. In our implementation we use DBSCAN [18], with $t$ as the noise tolerance and a minimum neighbourhood size 1 for core points. When splitting a cluster into cliques, if its largest range across a single dimension is $n > 1$ times the threshold $t$, it is split into $\lceil n \rceil$ clusters across that dimension. In the determinism refinement, when splitting a cluster $c$, there may be some $p \in c$ for which $p \cdot \sigma \notin P$. In this case a best match $c_\sigma$ for $\mathcal{O}_S(p \cdot \sigma)$ is found by the heuristic given in section 4.2, and $p$ is added to the respective new cluster.

## D   Synthetic Grammars

### D.1   Tomita Grammars

We adapt the Tomita grammars [33] for use as weighted models as follows: for each Tomita grammar and its minimal DFA $T$ we create a PDFA variant $T_W$ which has the same structure as $T$, and in which accepting/rejecting states are differentiated by their preference for 0 or 1. Every state in $T_W$ has stopping probability $0.05$, the states $q$ have transition weights $0.7 \cdot 0.95 = 0.665$ and $0.3 \cdot 0.95 = 0.285$, such that $\delta_W(q, 0) = 0.665$ iff $q$ is an accepting state in $T$. We show all of the adaptations in D.1, labelling the weighted variants T1 through T7 in the same order as their binary counterparts. The images were generated using graphviz.

We train 7 RNNs on these grammars, their parameters and training routine are described in E. We extract from them with the same algorithms as for the SPiCe and UHL languages. The extraction parameters and results are given in table 3.

From each of the Tomita RNNs, our algorithm successfully reconstructs a PDFA with the exact same structure as the RNN's target PDFA, and transition weights within tolerance of the corresponding weights in the target. The extracted PDFAs for each Tomita RNN are presented in D.2.

### D.2   Unbounded History Languages

The UHLs are 3 cyclic PDFAs, shown in D.3. UHL 3 is a weighted adaptation of Tomita 5, where the difference in probabilities between the states is lower than in our original adaptations. This makes it harder for the $n$-gram to guess the current state from local clues in its window (such as many appearances of one token over another). Precisely:

> UHL1 is a 9-state cycle PDFA over $\Sigma = \{0, 1\}$ that loops through all of its states one at a time, regardless of the actual input token. On all states it has stopping probability $0.05$, and divides the remaining next-token distribution over 0 and 1 as follows: on all states 0 has next-token probability $0.75$ and 1 has $0.15$, except for the second, fifth, and ninth states, where this is reversed.

> UHL2 is a 5-state cycle PDFA over $\Sigma = \{0, 1, 2, 3, 4\}$, that loops through all of its states one at a time regardless of input token. At every state it has stopping probability $0.045$, and it gives next-token probability $0.591$ to a different token at each state, with the rest of the tokens getting a uniform distribution between themselves.

> UHL3 is a 4-state PDFA over $\Sigma = \{0, 1\}$ that maintains the parity of the seen 0 and 1 tokens. Every state has stopping probability $0.05$, and most states give 0 next-token probability

Figure D.1: Weighted variants of the Tomita grammars.

Figure D.2: PDFAs extracted using WL* from the RNNs trained on weighted variants of the Tomita grammars.

| Language ($\|\Sigma\|, \ell$) | Model | WER↓ | NDCG↑ | Time (s) | WER Size | NDCG Size |
|---|---|---|---|---|---|---|
| Tomita 1 (2, 0.77) | WL* | **0.0** | **1.0** | 55 | 2 | 2 |
| | Spectral | **0.0** | **1.0** | 18 | k=10 | k=10 |
| | N-Gram | 0.0001 | 0.9998 | 27 | 63 (n=6) | 31 (n=5) |
| | ALERGIA | **0.0** | **1.0** | 28 | 8 | 8 |
| Tomita 2 (2, 0.78) | WL* | **0.0** | **1.0** | 55 | 3 | 3 |
| | Spectral | **0.0** | **1.0** | 13 | k=10 | k=10 |
| | N-Gram | 0.0 | **1.0** | 27 | 63 (n=6) | 15 (n=4) |
| | ALERGIA | **0.0** | **1.0** | 28 | 6 | 6 |
| Tomita 3 (2, 0.78) | WL* | **0.0** | **1.0** | 62 | 5 | 5 |
| | Spectral | 0.0071 | 0.9945 | 13 | k=7 | k=13 |
| | N-Gram | 0.0542 | 0.9918 | 27 | 63 (n=6) | 63 (n=6) |
| | ALERGIA | 0.0318 | 0.9963 | 28 | 8 | 8 |
| Tomita 4 (2, 0.79) | WL* | **0.0** | **1.0** | 56 | 4 | 4 |
| | Spectral | **0.0** | **1.0** | 13 | k=14 | k=12 |
| | N-Gram | 0.073 | 0.9887 | 27 | 63 (n=6) | 63 (n=6) |
| | ALERGIA | **0.0** | **1.0** | 28 | 9 | 9 |
| Tomita 5 (2, 0.79) | WL* | **0.0** | **1.0** | 56 | 4 | 4 |
| | Spectral | 0.0001 | **1.0** | 11 | k=67 | k=23 |
| | N-Gram | 0.1578 | 0.9755 | 27 | 63 (n=6) | 63 (n=6) |
| | ALERGIA | 0.0315 | 0.991 | 29 | 15 | 15 |
| Tomita 6 (2, 0.78) | WL* | **0.0** | **1.0** | 56 | 3 | 3 |
| | Spectral | 0.0003 | 0.9999 | 23 | k=36 | k=36 |
| | N-Gram | 0.1645 | 0.9695 | 27 | 63 (n=6) | 63 (n=6) |
| | ALERGIA | 0.0448 | 0.9983 | 28 | 12 | 12 |
| Tomita 7 (2, 0.78) | WL* | **0.0** | **1.0** | 63 | 5 | 5 |
| | Spectral | 0.0003 | 0.9999 | 13 | k=32 | k=37 |
| | N-Gram | 0.0771 | 0.9857 | 27 | 63 (n=6) | 63 (n=6) |
| | ALERGIA | 0.0363 | 0.9936 | 28 | 11 | 11 |

Table 3: Tomita results. Each language is listed with its alphabet size $|\Sigma|$ and RNN test loss $\ell$. The $n$-grams and sample-based PDFAs were created from 50,000 samples, and shared samples. FLEXFRINGE was run with state_count = 50 . Our algorithm was run with $t=0.1, \varepsilon_P, \varepsilon_S=0, |P|\leq 5000$ and $|S|\leq 100$, and spectral with $|P|, |S|=100$.

0.525 and 1 next-token probability 0.425, except for the state where the number of seen 0s and 1s is odd, where this is reversed.

UHL3 is an adaptation of the fifth Tomita grammar similar to our other presented adaptations, except that here the next-token probabilities of 1 and 0 are closer to each other, making it slightly harder to infer which states the PDFA has been in from a finite history[4]

Applied with variation tolerance $t = 0.1$, our algorithm managed to reconstruct every UHLs structure from its trained RNN perfectly, with weights within $t$ of the original[5]. The reconstructed PDFAs are shown in D.4.

# E   RNNs

All the RNNs are 2-layer pytorch LSTMs with training dropout 0.5 and linear transformation + softmax for the classification. The input token embeddings and initial hidden states were treated as parameters.

The Tomita and UHL RNNs had input (embedding) dimension 2 and hidden dimension 50, except for UHL 2 which had input dimension 5. The SPiCe RNNs had input/hidden dimensions (resp.) as follows: 0. 4/50 1. 20/50 2. 10/50 3. 10/50 4. 33/100 6. 60/100 7. 20/50 9. 11/100 10. 10/20 14. 27/30 .

Figure D.3: The UHL PDFAs.

The RNNs were trained with the ADAM optimiser and varying learning rates, each training for 10 full epochs for learning rate (or less if the validation loss stopped decreasing). The SPiCe and UHL RNNs used a cyclic learning rate, going through 8 values from 0.01 to 0.0001 2 and a half times. The Tomita RNNs simply used the learning rates $0.01, 0.008, 0.006, 0.004, 0.002, 0.001, 0.0005, 0.0001, 5e-05$ once in order.

The SPiCe RNNs were trained with the train samples given by the spice competition [7]. For the UHL and Tomita RNNs, we generated train sets of size $10,000$ and $20,000$ respectively by sampling from the target PDFAs according to their distributions. For each RNN, we split its given train set into train, validation, and test sets, taking respectively $90\%/5\%/5\%$ of the original set. We checked each RNN's validation loss after every epoch. Whenever it worsened for 2 consecutive epochs, we reverted to the previous best RNN (by validation loss) and moved to the next learning rate.

For each RNN, in each training epoch we randomly split the train set into batches of equal size (up to the last 'leftover' batch), and trained in these batches. For the UHL and Tomita RNNs we trained with batch size $500$ and for the SPiCe RNNs we used $1,000$.

Figure D.4: The UHL PDFAs, as reconstructed by WL*from RNNs trained on the original UHLs.

## Footnotes

[4]This recalls the insight of [31], who note that unexpected tokens are useful as they convey information about the current state of the model.

[5](When extracting from RNNs, the weights of course can only be as good as those learned by the RNNs)