[Reviews · NeurIPS 2019]

Reviewer 1



They introduce an algorithm for inferring "probabilistic deterministic finite automatons" from a probabilistic language model oracle that can compute P(next_symbol | string_so_far), which is exactly what a autoregressive neural language model, like a rnn, would give you. The paper is well-written, the problem is well motivated, and the algorithm appears to be better than the prior work, as measured by the experiments. However I am not qualified to adequately review this paper, because I'm not familiar with any of the related work. For this reason, I lean toward accepting the paper, but I have very little confidence in my assessment. The only problems I have with this paper are with the experiments. They evaluate on SPICE data sets, which appear to be standard data sets for finite state language model learning. However, in looking at the cited SPICE paper, there are actually 16 SPICE data sets, and they only evaluate on 4 of them. Why only these four? Are the other twelve too hard? It would be okay if your algorithm cannot handle the other twelve, provided that the prior work also does poorly on this test cases. Relatedly, what would happen if you apply your algorithm to a language model trained on a large natural language corpus? I imagine that this is an ultimate intended use case of distilling finite state machines from neural language models, so that for example you could use the (finite-state) language model on low power or embedded devices where floating-point operations are at a premium. In a similar vein, state-of-the-art language models for natural language are based on transformers, and as far as I can tell your algorithm would work equally well with a transformer model, or indeed any autoregressive model. This might be an interesting experiment to do, or at least mention as a possibility. "Probabilistic deterministic" finite state machines sounds like an oxymoron, but really what is meant are probabilistic symbol emissions and deterministic state transitions. For readers like me that are not familiar with this literature, a quick sentence in the introduction clarifying this would be helpful (the third paragraph and lines 112-115 make the meaning clear, but it would be helpful if it came a bit earlier). The equation on line 125 might have a typo in it: $\sigma$ appears as a free variable in the left-hand side, but does not occur in the right hand side. Did you mean $... = \frac{P^p(w\sigma)}{P^p(w)}$ ? This would make sense for calculating the probability of $\sigma$ following $w$.

Reviewer 2



I think the paper is in a very interesting area but feels unfinished. The paper is reasonably well structured, though some parts felt repetitive (e.g. lines 45-75 in the introduction). Some definitions were difficult to find (typos + small comments are at the end of this section), but overall the descriptions of the background and model were easy to follow. Ultimately, the experiments appear inconclusive. The paper only shows results for SpiCe challenge 0-3, where challenge 1-3 are synthetic examples (simplest) and challenge 0 were not a part of the competition scoring. So, it is unclear how the model will scale. The Tomita experiments were also small and inconclusive. It is also unclear how the RNN hidden size is chosen and how it will affect the WL and WFA algorithms. It would also be helpful to understand how different hyperparameters of the model affect results. The variation tolerance $t$ matters a lot. It would be interesting to understand why t=0.1 is chosen, and what the effect is in altering $t$ for all experiments. The table expansion time and the random seed probably also affects performance. I am curious about some design choices. For example, what is the significance of using total variation vs other distance measures (e.g. KL divergence or cross-entropy)? Also, in the experiments, it’s not clear whether Step 3 (answering equivalence query) is ever used. What should be done if the hypothesis is not accepted? Typos and small comments: * The notation $w^k$ in line 108 should be (I think) $w_{:k}$ * The colon in line 28 should be a comma * I didn’t see an explanation for what the contents of the observation table $O$ is -- with the exception of lines 182-183, which states that the observations are tokens and not distributions. But the following line 184, along with the rest of the paper, discusses the comparison between *next-token distributions*. (Are you using RNN prediction probabilities for comparison or the empirical probabilities from the samples?) * The notation $q^i$ doesn’t appear to be defined. I assume it is referring to the initial state. Maybe use the notation $q_0$ instead, to since $i$ is often used as an index? * In line 125, the definition of the last token probability, the equation should be $\frac{P^p(w \cdot s)}{P^p(w)} * In line 193, “prefix weight” does not seem to be defined * The Tomita experiment results were difficult to find. Maybe a subsection heading or a simple figure would help

Reviewer 3



This paper presents a technique for constructing a probabilistic deterministic finite automaton (PDFA) that can model a black-box language model such as LM-RNN. The main idea of the algorithm is to adapt Angulin’s L* algorithm to handle probabilistic choices with unbounded states of an LM-RNN by developing a notion of variation tolerance. The variation tolerance allows for comparing two probability vectors and clustering them if they are within a t-threshold. The goal is to construct a PDFA, such that for all prefixes the next token-distributions in the PDFA and the LM-RNN is within the variation bound. The paper presents analogous variation tolerance aware extensions to membership and equivalence queries in L*. The algorithm first learns an observation table that is closed and consistent, then constructs the corresponding PDFA using a clustering strategy, and finally performs an equivalence query using a sampling-based method. The technique is evaluated on grammars from the SPiCe competition and adaptations of Tomita grammars, and it outperforms n-gram and WFA (using spectral learning) baselines both in terms of WER and NDCG rates. This paper presents a novel general technique to learn weighted deterministic finite automaton (WDFA) from a given weighted black-box target, and learns a PDFA for a stochastic target. Unlike previous approaches that use spectral learning or learn from sampled sequences, the presented technique learns PDFA in an active learning setting using an oracle by extending the widely used L* algorithm, which has some nice guarantees. The idea of using t-consistency for computing variations between probability vectors is quite elegant, and the idea of using clustering techniques to overcome non-transitivity of t-equality is also quite interesting. There is also a detailed formal treatment of the properties and guarantees of the returned PDFA in terms of t-consistency. The paper is also nicely written and explains the key ideas of the algorithm in required details. One suggestion would be to add a small running example that might help clarify the extended L* algorithm a bit more (especially the step 2 of PDFA construction). It would also be nice to move at least the two theorems from appendix to the main text. I was curious about the scalability of the algorithm. It seems it scales better than spectral learning based WFA learning methods based on SPiCe and Tomita benchmarks. But the LSTM network consists of an input dimension of 10 and hidden dimension of 20, which seems quite small compared to current state-of-the-art LSTM language models. What is the maximum hidden sizes the technique can handle currently? Similarly, the alphabet size of upto 20 also seems a bit small compared to typical large vocabularies of state-of-the-art language models. It would be helpful to report the maximum sizes the technique can scale to currently, and that might help spur interesting future works to scale up the technique further. It wasn’t clear whether for the results reported in Table 1, did the L* algorithm terminate before early stopping. If it didn’t terminate, what bounds on expansion time and suffix limits might be needed for full completion of L* algorithm on these benchmarks. Since the equivalence check is being performed using sampling, the only difference between previous PDFA learning methods from samples would be that in this case the samples are being collected actively. Would it be possible to compare the presented L* based technique to also some of those previous PDFA reconstruction techniques that learn from samples? Minor issues: line 108: w^k \in S -> w_{:k} \in S line 165: algorithm reduce the number -> algorithm reduces the number line 193: prefix weight: -> prefix weight. line 336: The the -> The

[Author Response · NeurIPS 2019]

We thank the reviewers for their detailed and insightful reviews. We will incorporate all presentation changes as suggested. Below we answer the main questions raised by the reviewers.

**Evaluation on Larger Grammars (all reviewers)** The reviewers note that the results section in the submitted paper appears inconclusive, or that the experiments have been run on very small languages and with small alphabets. Sadly, the algorithm is indeed not currently applicable to large 'complicated' languages, although (as we present below) it is generally more successful in these attempts than the spectral algorithm. We note that the results in the paper do show that on the synthetic SPiCe and Tomita grammars, the L*-learned PDFA (and occasionally spectral-learned WFA) outperform n-gram, presumably because of the ability of finite state machines to capture patterns that an n-gram is unable to encode. We believe that while the algorithm is not scalable to large grammars, it is an interesting new step into the field of WFA extraction, and worth sharing with the community. We hope that in the future others will be able to work on and expand this algorithm so that it is useful even for natural languages.

Since the original submission, we have managed to train and extract from networks for some more complicated SPiCe grammars, in particular: SPiCe 4 (NLP), 7,10 (biology), and 6,9,14 (part-synthetic), which have alphabet sizes ranging from 11 to 60, we present these new results now. For each language we trained a network with 2 or 3 layers, and hidden dimension of size 20-100 (depending on the language). Unless stated otherwise, the hyper-parameters of the extraction were: L* with maximum $|P|$=5k and variation tolerance $t = 0.1$, spectral with a Hankel matrix of size 500x500, and n-grams with total sample length 5 million and $n \in [6]$. The extracted models' WER against their targets and extraction time are provided in the table below. Our approach outperforms spectral extraction on all but one benchmark (SPiCe 9). For SPiCe 10, allowing spectral to expand to Hankel 1000x1000 remains at WER 0.863, and takes 1.8hrs. Interestingly, the best WFAs for SPiCe 10 were always those with $k = 1$.

| Name | Our Approach | Spectral | n-gram |
|------|------|------|------|
| SPiCe 4 | 0.318 (1.8hrs) | 0.348 (1.8hrs, 1000x1000) | 0.112 (0.9hrs) |
| SPiCe 6 | 0.575 (2.5hrs) | 0.788 (1.4hrs), 0.682 (6.1hrs, 1000x1000) | 0.274 (0.8hrs) |
| SPiCe 7 | 0.625 (0.5hrs) | 0.801 (0.6hrs) | 0.442 (0.7hrs) |
| SPiCe 9 | 0.485 (0.5hrs) | 0.287 (0.4hrs) | 0.116 (1hr) |
| SPiCe 10 | 0.646 (0.9hrs) | 0.865 (0.4hrs), 0.863 (1.8hrs, 1000x1000) | 0.347 (0.8hrs) |
| SPiCe 14 | 0.329 (1.3hrs, |P|=10k) | 0.612 (1.6hrs) | 0.075 (0.8hrs) |

Table 1: Word error rate (WER) of extracted models for larger languages. Our approach outperforms spectral extraction in all but one benchmark (Spice 9).

**Effect of Hidden Size of Networks on Extraction (rev4, rev5)** All of the algorithms evaluated in the paper are agnostic to the internal structure of the language model under extraction (in our case, an RNN), and in particular to the size of its internal state (i.e. hidden size for RNNs). Note that the experiments presented above have networks with larger hidden size (20-100) than shown in the paper, to allow learning of the more complicated languages.

**Choice of Hyper-parameters, and their effect of Hyper-parameters on Results (rev4)** For the variation tolerance parameter, the original heuristic was to set $t = 1/|\Sigma|$. The intuition for this was that given no data at all, the fairest distribution one can give to tokens is the uniform distribution, and so this may also be considered the threshold for whether a token is deemed 'likely' by a given model or not. From this we extrapolate that a reasonable threshold for significant difference between the probabilities of two tokens is also the uniform probability, though for larger alphabets we may quickly change this to $1/n$, where $n$ is an estimate of how many tokens are generally likely after any given prefix. In practice, we see in the examples above that using $t = 0.1$ already strongly differentiates even models with larger alphabets (these extractions did not reach equivalence), and so did not use smaller $t$. Starting out with a large $t$, and then reducing it so long as the model is reaching equivalence quickly, would also be a good strategy. We will more carefully research the effect of this parameter on the extraction, and add a fuller discussion and evaluation of all hyper-parameters to the paper.

**Use of Equivalence Queries and Handling of Counterexamples (rev4)** Reviewer 4 notes that it is unclear whether step 3 (equivalence queries) of the algorithm is ever used. This is a good question considering the results presented in the original submission, which did not discuss counterexamples. We note that in the extractions presented above, possibly following the addition of the thresholding for new prefixes and suffixes, the equivalence query is invoked often and successfully, regularly rejecting hypotheses. In this case, all of the prefixes of the returned counterexample are added to P, and the observation table is expanded until it is again closed and consistent. We will clarify the relevance of this step in our work by recording the number of counterexamples returned during each extraction.

**Comparison to Sample-Based PDFA Reconstruction Techniques (rev5)** We have not had time to prepare a comparison to these techniques for the response, but we agree that this is an important comparison to make, and will add an evaluation of one such technique to our results in the final version of the paper.

[Meta-Review · NeurIPS 2019]

This paper make substantial progress on a well-motivated methodological problem in a theoretically principled way. The methods appear not yet to be practical, but there are substantial novel ideas, and a rigorous analysis. This work appears likely to serve as a springboard for future research.